# Comparing migration of Whinchats *Saxicola rubetra* from the non-breeding grounds in Liberia and Nigeria: Differences due to geography but otherwise very similar

Will Cresswell[1,2]*, Rob Patchett[1], Emma Blackburn[3], Malcolm Burgess[4]

**1** Centre for Biological Diversity, University of St Andrews, Harold Mitchell Building, St Andrews, Fife, United Kingdom, **2** A.P. Leventis Ornithological Research Institute, Jos, Nigeria, **3** Paloverde Dr, Loveland, Colorado, United States of America, **4** RSPB Centre for Conservation Science, The Lodge, Sandy, Beds, United Kingdom

* wrlc@st-and.ac.uk

## Abstract

Migrant birds in the Afro-Palearctic region are declining, so understanding general migration characteristics, such as site use, connectivity, and phenology is crucial for their conservation. We tracked 64 whinchats *Saxicola rubetra*, a declining Palearctic-breeding passerine, from non-breeding sites in Nigeria and Liberia, to Europe and back, in multiple years. We predicted differences, resulting from the geographical location of the two non-breeding sites, in location of respective breeding areas (migratory connectivity), number of non-breeding and stopover sites, migration distance and duration, degree of loop migration and phenology. But we predicted similarities, resulting from optimising migration behaviour, in migration leg distance duration, and stopover duration. Liberian tagged birds bred mainly in central and northern Europe, with Nigerian birds mainly in eastern Europe. Migratory spread was large resulting in range overlap and low connectivity. About 25% of whinchats used more than one sub-Saharan non-breeding site, their location dependent on geographic availability. Liberian birds had longer migrations in distance and duration, and more stopovers, but only Nigerian birds showed a statistically significant difference in longitude comparing spring and autumn migrations (i.e., a clear loop migration). Nigerian birds departed later than Liberian birds, independent of breeding latitude or migration distance, although latitude determined arrival time for breeding. Spring migration leg distance and stopover duration, and all leg durations, were similar for both populations; but Nigerian birds had longer duration stopovers and shorter distance migration legs in autumn. Whinchats were shown to have varied migration routes and characteristics, with a variable pace of migration that allows them between 50 and 75% of daylight hours to rest and forage, but with major stopover duration possibly being affected by site quality. One whinchat moved from sub-Saharan non-breeding site to northern European breeding site in 7 days, and 8% of

**Data availability statement:** Data relevant to this study are available from the University of St Andrews' PURE data depository at https://doi.org/10.17630/59c8ff25-858d-4dcb-8f47-6b65894bd0e0.

**Funding:** Chris Goodwin, A.P. Leventis Conservation Foundation, AP Leventis Ornithological Research Institute, the British Ornithologists' Union and the Linnean Society The funders had no role in study design, data collection and analysis, decision to publish, or preparation of the manuscript

**Competing interests:** The authors have declared that no competing interests exist.

complete migration durations were 14 days or less. This suggests whinchats are well adapted to the current variable geography and so may have the capacity to adapt to potential climate change across Europe and West Africa, although average quality and availability of stopover sites may be contributing to declines.

## Introduction

Migrant birds are declining globally [1–3] and populations of many Afro-Palearctic long-distance migrant bird species, across all functional groups, have been declining for the last 50 years [4]. That so many different species of migrants are declining, with very different ecological niches, suggests that it is the general characteristics of migration that make them particularly vulnerable to anthropogenic climate and habitat change [5]. Migrant birds need a range of suitable sites over a wide geographic area throughout their annual cycle [6]: removal or change at any one of these sites can reduce survival and productivity [7]. Nevertheless, the general characteristics of most migrants – their low connectivity [8], serial residency [9] and their tracking of optimal climatic conditions in time and space [10] should potentially make them resilient to such changes in their non-breeding areas.

Understanding migration characteristics, such as connectivity, site use, migration legs and stopovers, and phenology is therefore crucial for effective conservation management of declining species [5]. Some migration characteristics should be similar across all populations of a species, reflecting adaptive optimisation, particularly during spring migration [11]. Migration is costly relative to the stationary periods [e.g., [12], [13]], with high survival particularly during the non-breeding period [e.g., [14], e.g., [15]], so the number and duration of stopovers should always be reduced, along with migration distance, in most situations. But there are a number of important modifiers to this general pattern. Geography (i.e., the spatial configuration, type and distribution of land) will influence choice of non-breeding locations [8], and their distance from the breeding site as well as availability of suitable habitat for refuelling [e.g., 6, 16]. Wind strength and direction will influence energy required for flight [17]. Temperature will influence habitat suitability on arrival at the breeding grounds [e.g., [18], [19]]. All three may then independently, or in concert, constrain routes and site use, and so connectivity; the use of optimal leg lengths and stopover periods; and timing during migration.

Migration routes and destinations have been shown to be highly variable between individuals on first migrations, with the majority of species having low connectivity [8], arising from high migratory spread [20]. Because of the variable and unpredictable climate on a decadal basis within the main non-breeding areas for Palearctic migrants in Africa [21], most migrants show a bet hedging strategy of low connectivity [the serial residency hypothesis – see 9]. Some juveniles arrive in suitable areas, and the survivors return to these suitable areas in subsequent years; but as climate changes, new suitable non-breeding areas are successfully reached and colonised by some juveniles each year [e.g., 22]. Consequently, low connectivity should be

conserved in most populations of migrants, with corresponding lack of selection for habitat specialisation, allowing dispersing juveniles and indeed adults during unfavourable migration events to be able to survive in the wide range of habitats they must encounter [9,23]. This means that most migrant species have potentially wide non-breeding ranges and so a wide range of starting locations [20], and so different geography to deal with on migration. There should therefore be considerable variation in migration distance and routes across widely distributed populations of migrants, and some degree of overlap in breeding areas, even for spatially separated non-breeding populations.

Geography may also increase variability in non-breeding location and use of multiple non-breeding sites prior to migration [3]. Migrants should choose areas with suitable habitat for their non-breeding site that is closest to the breeding site, again to adaptively minimise migration costs [24]. Areas of Africa used by non-breeding Palearctic migrants vary in geography and so in the extent of suitable climatic zones and habitats [25]. There should therefore be more variability in the number and spread of non-breeding sites used by populations of migrants moving eastwards across Africa, where climatic zones become larger and land that might contain suitable habitat is much more available.

Geography may reduce variability, however. The major geographical barriers to migration on the Afro-Palearctic flyway, the Sahara desert and the Mediterranean sea, extend across the entire width of Africa, except at the extreme east and west for the Mediterranean [6,25]. Similarity in occurrence of areas where habitat is unsuitable (i.e., barriers where stopovers and/or foraging is impossible) across the range should produce similar optima that are conserved across populations. Non-stop flights [in the sense of flights without refuelling or rest stopovers longer than a few hours every day – see 26] across the Sahara and/or the Mediterranean require maximal fuel loads that require a sufficiently long pre-migration stationary period to acquire [27]. Maximal fuel loads then allow a migrant to maximise flight range and to minimise the number of stopovers, and/or to survive unfavourable weather that might slow migration [28]. There should therefore be similar migration leg length and duration, travel speed during migration legs and stopover duration across populations of migrants.

Climate and patterns of temperature [24] and wind [29] will both create similarities and differences in migration characteristics across migrant populations. To an extent, migration is an adaptation of species to reduce exposure to climate variation – migrants follow optimal climatic conditions [30,31]. There should therefore be similarities in stopover durations and use of non-breeding areas across populations. The timing of optimal climatic conditions, however, varies, particularly on the breeding grounds [32,33]. Arctic or montane breeding populations may only find suitable conditions for breeding several weeks later than those breeding in temperate areas [e.g., 34]. There may therefore be variation in the phenology and overall duration of migration across populations dependent on starting non-breeding area and finishing breeding area.

Although climate varies seasonally (with an increasing rate of change due to anthropogenic climate change) there is a predictable interaction with geography, where the prevailing wind direction during spring or autumn migration varies dependent on non-breeding and breeding location [35]. Optimal routes in the sense of energy minimisation, rather than distance minimisation, may then vary across populations as they utilise wind assistance, or avoid consistent head winds, leading to loop migrations in some populations [36].

Understanding these migration characteristics is important to understand why migrant species are declining. Here we tracked 64 whinchat *Saxicola rubetra*, a widespread small landbird of least conservation concern, but with substantial declines in Western and Central Europe [37,38], over their annual migration cycle from non-breeding sites in Nigeria and Liberia (separated by 2,100 km), to Europe and back, across multiple years. Note that some of the tracks from the birds providing data in this analysis have been used previously for a limited analysis of spring migration barrier crossing [n = 35 birds, [39]] and connectivity variation between years [n = 29 birds, 40], but here we reanalyse all tracks exactly in the same way, and combine them with a larger sample size from multiple studies throughout their annual cycle, so that they are comparative.

We tested for differences in migration characteristics between the two country non-breeding populations that might arise because of the difference in their locations, and consequent difference in the availability of land for breeding and

 

staging directly to the north, and non-breeding sites to the south-west. Liberian birds occur at the far southwest of West Africa, with limited availability of land to the south and west and potential breeding areas only to the northeast, rather than directly north; Nigerian birds, effectively in central Africa, have no such constraints (see Fig 1). Because of the problems in distinguishing between extended stationary stopover periods and use of additional sub-Saharan non-breeding sites in the spring we compared migration characteristics with and without any equivocal sub-Saharan stationary periods. We predicted that:

1. Liberian birds will use breeding areas in Northern and Western Europe compared to Central and Eastern Europe for Nigerian birds and so there will be limited – but some –overlap in breeding range because of low migratory connectivity and high migratory spread.

2. Liberian birds will use fewer non-breeding sites to the south of their main non-breeding site than Nigerian birds.

3. Liberian birds will have longer total migration distances, more stopovers and much less of a loop migration, if at all.

4. Overall migration duration and phenology will vary between birds from Nigeria and Liberia, but will be driven by population differences in overall migration distance and breeding latitude.

We also tested for similarities in migration characteristics that might arise from optimising migration behaviour within whinchat's physiological constraints, and geographical barriers (the Sahara and the Mediterranean). We predicted that Liberian and Nigerian whinchats will have similar:

5. migration leg distance and migration leg duration (and therefore travel speed during a migration leg), and stopover duration.

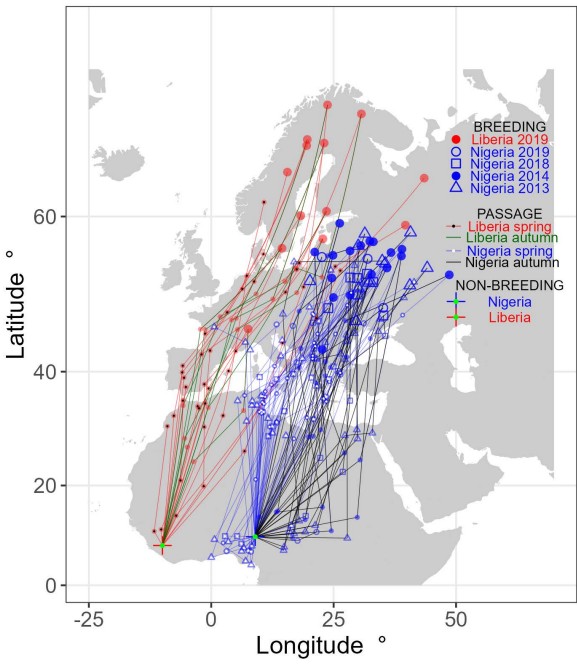

**Fig 1. The stationary periods (greater or equal to 2 days) of 64 whinchats tracked with geolocators from their primary non-breeding grounds in Nigeria (blue) and Liberia (red) in several years.**

## Materials and methods

### Study system

Whinchats were tagged in Nigeria and Liberia in multiple years. Whinchats were tagged on the Jos Plateau in the guinea savannah zone of central Nigeria, West Africa (9.8659N, 8.9686E, approximately 1250 m altitude) and in north-east Liberia at Wologizi (Alabama Camp), in a large patch of derived savannah in the rain forest zone of West Africa (8.1190N −9.9760E, approximately 540 m altitude). The sites are separated by approximately 2,100 km. Whinchats were tagged and tracked from Nigeria between February 2013 and November 2013 (study 1); February 2014 until April 2015 (study 2); November 2017 until November 2018 (study 3); November 2018 until November 2019 (study 4). Whinchats were tagged and tracked from Liberia from January 2019 until January 2020 (study 5). Whinchats were captured in Nigeria within an area of approximately 40 km$^2$; full site details are described in Blackburn and Cresswell [41]. Capture areas were principally open scrubland with varying degrees of habitat degradation from human habitation, arable farming and livestock grazing, the latter increasing in intensity over the dry season [see 42, 43]. Whinchats were captured in Liberia within an area of approximately 12 km$^2$. Capture areas were open savannah with much lower human population density, few agricultural fields and little cattle grazing compared to Nigeria, but both study areas represented typical non-breeding habitat for this species in the region (open, derived savannah, with a mosaic of small fields and nomadic grazing) and had a high density of Whinchats.

Whinchats in this study were caught with spring traps and mist nets. Birds were aged and sexed [44], ringed with unique combinations of colour-rings, and fitted with a geolocator. We deployed 49, 94, 60, 49 and 73 geolocators; recovered 17, 22, 8, 8, and 19 geolocators from which we downloaded 16, 19, 8, 5, and 16 usable tracks in studies 1–5 respectively. Geolocators were of various types and design, but predominantly Lotek UK MK6740 tags with 10 mm (n = 96) or 5 mm (n = 47) angled light stalks, but also 34 ML6540 tags without light stalks for studies 1 and 2. Full details are given in Blackburn *et al.* [45], including a full analysis demonstrating that the different designs of tags did not significantly affect survival, although there was a suggestion that 5 mm light stalks had a lower effect on survival than 10 mm light stalks. Further analysis of data from studies 1–3 [46] showed that 10 mm light stalks did not improve precision or accuracy of geolocation data compared to 5 mm stalks. Therefore, for studies 3–5, we again used Lotec UK MK6470 tags but with 5 mm light stalks.

Tags were fitted using leg-loop 'Rappole-Tipton' (also called backpack) harnesses. Full details of tag and harness design are given in Blackburn *et al.* [45]. Tags weighed on average 0.63 g (± 0.01 SE N = 156), representing 4.1% (± 0.05 SE) of average body mass [45]. Attempts were made to recapture any returning tagged bird resighted in the following winter; one individual was captured two years later and its logger recovered. Upon recapture, geolocators were removed by cutting the harness and birds were released unharmed after briefly assessing body condition [see 45]. Only one complete migration was used for each individual in this paper (i.e., no repeated tracks).

The effects of tagging on survival of whinchats was fully evaluated in studies 1 and 2 [see 45]. In summary, in study 1, of 37 whinchats fitted only with colour rings as control, 16 (43.2%) were resighted one year later compared to 10 (38.4%) of 36 tagged birds. In study 2, of 279 whinchats fitted only with colour rings as control, 74 (26.5%) were resighted one year later compared to 39 (30.0%) of 130 tagged birds. Overall we could find no reasonable evidence that survival varied as a consequence of tagging in individual years or years pooled [45]. We further confirmed this by only fitting colour-rings to 43 whinchats as controls in study 4; 10 (23.3%) were resighted one year later compared to 10 (20.4%) of 49 tagged birds ($\chi^2_1$ = 25.9e-4, P = 0.9). We also further confirmed this by only fitting colour-rings to 32 whinchats as controls in study 5; 9 (28.1%) were resighted one year later compared to 19 (28.4%) of 66 tagged birds ($\chi^2_1$ = 2.6e-31, P = 1). Note sample sizes differ from overall number of tagged birds above because survival/resighting analyses only include tagged and control birds that were resighted on the study area afterwards during the period of tag fitting and so considered as resident rather than passage birds [see 45]; similar survival estimates in study 3 could not be used because residency of tagged and control birds was not established through sufficient resighting effort post tagging/colour-ringing. Overall, the average

difference in resighting rate for tagged versus control birds across the four studies was −2.8% (± 3.3 SE, N = 4 studies), and the overall percentages of birds resighted when pooling years were 27.8% for controls (N = 391) versus 27.8% for tagged birds (N = 281).

## Data processing

Raw data were downloaded, viewed and preliminarily cleaned using the BASTrack software suite [British Antarctic Survey, Cambridge, UK; see [47] for an overview of the following processes]. We adjusted for clock drift, assuming that any drift was linear. We used the Transedit2 software that is part of the BASTrack software to view raw data as light curves over time. We used a minimum observed light value of −1 to define sunrise and sunset, effectively treating any light record as a valid indicator of before sunset or after sunrise. We use the logic that although false negatives may be common (because a bird goes to roost early in or leaves late from a dark place), false positives are very unlikely because whinchats do not use habitats with artificial light. Detailed visual examination of individual light curves did not show any false positives during known dark periods at the tagging site. In practice, the first and last light record each day was used from each geo-locator and was used to provide one sunrise and sunset time for each day respectively.

Most analyses were then carried out using R 4.3.2 [48]. Stationary periods (stopovers of two or more days or breeding locations) were determined from visual inspection of plots of sunrise and sunset times with day. Note that stopover periods of less than two days, including daily stopovers of a few hours that may occur regularly during migration legs cannot be discerned from geolocator data except by a lower speed of travel within a leg than expected if a passerine was travelling continuously at an expected average flight speed (see Discussion section 5). A sudden and consistent change in sunrise and sunset indicated that a bird had changed location; a consistent series of similar sunrise and sunset times indicated a bird was stationary (see S1 Fig.). Each migration period was then identified in the data by labelling the start day and the end day of the migratory leg, and each consecutive stationary period, between migration legs, was given a unique identification number. We also visually analysed variation in sunrise and sunset times using another standard methodology for geolocators based on the R library FlightR [49,50]. Light and dark periods were visualised using the 'lightImage' function and the predicted twilights for the non-breeding, tagging location in Nigeria were identified [e.g., see Fig 2 in 46]. Locations were then adjusted to fit time of sunrise and sunset curves that fit the light data for other stationary periods, which then highlighted discontinuities (sudden changes) in sunrise and sunset, coincident with a migration stage, allowing the start and end of stationary periods to be easily identified. This method is particularly useful to identify birds that had breeding locations north of the Arctic Circle, and therefore some periods during mid-summer where daylight was 24 hours: these periods were then excluded from any calculation of location (i.e., only the start and end of the breeding period, when there was some period of dark available to calculate locations for Arctic birds). Both methods identified similar stationary periods.

Whinchat locations were then determined from the sunrise and sunset times within each unique stationary period. We used the R library Geolight to determine latitude and longitude at an accuracy of approximately 100–500 km, subject to assumptions of the sun elevation angle and to imprecision due to shading [51,52]. Sun elevation angle is unknown unless location is known (and vice versa), but a sun elevation angle must be used to calculate latitude (to determine effective sunrise). Here, we take the simple approach of using the sun elevation angle that correctly located the bird at the tagging and recapture site: the only place where location is certain. This uses the logic that although we may not know the correct sun elevation angle for a single bird away from the tagging site, we assume any individual has an equal chance of having a lower or higher sun elevation value than the original, known sun elevation angle, and so errors are normally distributed. Comparisons between birds are then subject to errors (imprecision) but not bias. If errors are normally distributed, the use of a single sun elevation angle assumption prevents systematic bias, so that averaging across individuals should give true means with respect to distance travelled and speed, even if there is uncertainty about where any individual started or finished its migration leg. Previous analyses varying sun elevation angle have shown that location variation at the

geographic scale of the analyses carried out here do not greatly change the results, when hypotheses tested are comparing populations of geolocator tagged birds, rather than attempting to find precise locations of their migratory stopovers [see 39, 46]. Furthermore, errors that arise from variation in light detection by the tag and using an incorrect sun elevation angle are relatively small in Europe compared to sub-Saharan Africa where variation in day length is much smaller [40,45,46].

Geolocator locations are also subject to imprecision calculating latitude at the equinox periods (the few days either side of March 20th and September 22nd when regardless of latitude, daylength is similar). Although latitude is unreliable during these periods, estimation of longitude, that depends on sunrise and sunset times, remains consistently accurate. Assessment of migration phenology also remains consistently accurate, except in the very rare cases where a migration leg both occurs close to the equinox and broadly follows a north-south axis, without any longitudinal change so that both sunrise and sunset times would not be seen to change on any visual examination of the data. There were 16 migration periods that occurred within one week of the equinox and the mean change in longitude was 9.3 degrees, with a minimum value of 4.2 degrees. In terms of change of sunrise and sunset times, both would shift by at least approximately 17 minutes for the minimum change in longitude we observed, large enough to be detected by graphical inspection (see S1 Fig.), and so allowing the migration period to be identified, if not the accurate latitude of the stationary periods either side of it. In any case most of the migration periods within a week of the equinox (10/16) involved sunrise or sunset shifts of more than one hour. Furthermore, only a few of the hypotheses tested in this paper are affected by this imprecision in calculating latitude caused by migration close to equinox (see Prediction 5 below), and in these cases, analyses were also run excluding the data from the equinox periods to determine if there was any effect of the increased uncertainty in the data during these periods.

The resulting final data file then consisted of 362 rows, one for every migration period for each tagged bird, with the start and end, date, latitude and longitude, of the migration stage, the distance and duration between the start and end of each stage, along with individual tagged bird identity and country. The start location then represents the location of the prior stationary period, and the end location represents the location of the following stationary period. Great-circle distances between these start and end locations (i.e., migration distance between stationary periods) were calculated using the distHaversine function in the geosphere R library. The duration of each migratory leg and each stop-over period between them were then calculated from the start and end dates (to the nearest half day) of each stationary and migration period.

### Data analysis

Sample sizes were: 64 birds tracked, 16 from Liberia in 2019 and 48 from Nigeria (16 in 2013, 19 in 2014, 8 in 2018, 5 in 2019). Forty-one individuals provided complete tracks, from main non-breeding area to the breeding area and back again; a further 9 provided a complete track from main non-breeding area to the breeding area but the tag failed during breeding; a further 13 failed during spring migration between the main non-breeding area and the breeding area and so only provided limited data; a final 1 failed on the autumn migration between the breeding and non-breeding ground. There were 5.9 (± 1.5 SE) stationary periods for Liberia and 5.6 (± 0.8 SE) stationary periods for Nigeria recorded per tracked individual. Ages were similar in the samples across countries ($\chi^2 = 0.14$, P = 0.71): comparing 9 adults and 7 first year birds for Liberia, and 19 adults and 22 first year birds for Nigeria (a few birds could not be aged). Sexes were similar in the samples across countries ($\chi^2 = 0$, P = 1): comparing 6 females and 9 males for Liberia, and 16 females and 29 males for Nigeria (a few birds could not be sexed). Whinchats were broadly similar in their biometrics between countries: mass, tarsus length and pectoral score were similar, but with a trend for wing length to be longer in Nigerian birds (S1 Table). Age and sex were added to all models below but did not improve them in any statistical or biological way, and so these terms were not included in final models reported, to maximise statistical power (because of missing values in both age and sex).

Data were analysed using General Linear Mixed Models in R using the library nlme assuming a normal distribution; model fits were evaluated from diagnostic model plots, and assumptions were reasonably met in all models presented here [53]. Models all had the starting format of:

$$\text{Hypothesis variable} \sim country\,(Liberia/Nigeria) + season\,(spring/autumn) + country * season$$

A random effect of bird identity was included in every model to account for variable numbers of repeated measures per tagged bird where this was the case (i.e., when the initial sampling unit was a migration period, N = 362); some models use only one average or total value per bird (i.e., when the sampling unit was a tagged bird, maximum N = 64 but variable because some birds were not sampled over the whole annual cycle) and therefore do not include a random effect and were analysed using Generalised Linear Models. The interaction was removed if not significant and removal decreased AIC by at least 2 points.

**Prediction 1.** We predicted that Liberian birds would use breeding areas in Western Europe compared to Eastern Europe for Nigerian birds. We plotted stationary periods (or stopover locations, e.g., Supporting information Fig 1) using the raster, rgdal, rgeos and mapproj libraries in R (see Fig 1). Breeding range for birds from each country was calculated as the area contained within the minimum convex polygons of breeding locations, using the geosphere and alphahull libraries in R. Overlaps were calculated from the intersection of these polygons (see Fig 2). Migratory spread was calculated as the mean distance between all possible pairs of breeding sites for Liberia and Nigerian birds [see 8]. Confidence intervals were obtained by calculating the distance between 10,000 randomly selected pairs of breeding locations for each country, using the 250th and 9750th values as the upper and lower limits.

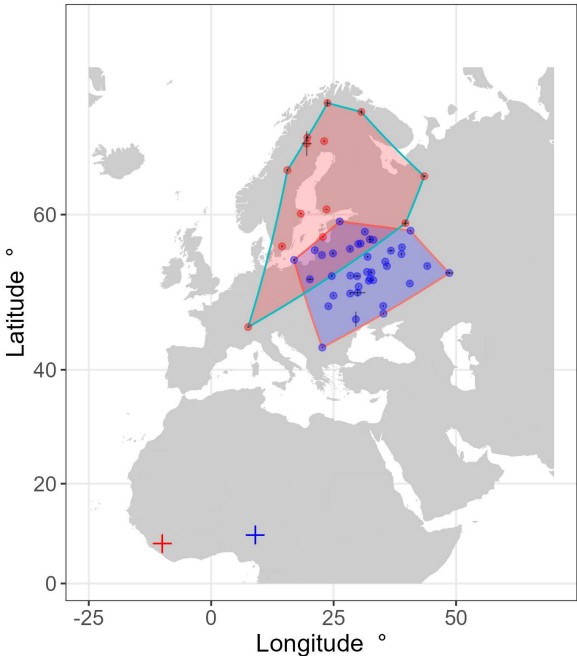

**Fig 2. The breeding range of Nigerian (blue shaded polygon) and Liberian (red shaded polygon) whinchats tracked with geolocators.** Minimum convex polygons using their breeding locations are plotted. Locations are plotted with one standard error of estimated latitude and longitude assuming the same sun elevation angle as measured on the main non-breeding site (most error bars are too small to be visible at this scale).

**Prediction 2.** We predicted that Liberian birds would use fewer subsequent non-breeding sites to the south of main non-breeding sites than Nigerian birds. We plotted all stationary periods south of the Sahara and away from the tagging sites (all were over 250 km away), of more than 2 days (range 13–52 days, apart from one of 3 days, representing a third non-breeding site, that was included because it followed a second non-breeding site that was occupied for 19 days, and both involved substantial movements in the opposite direction to subsequent migration to the breeding ground), as an additional non-breeding site (see Fig 3). Difference in the probability of having more than one sub-Saharan non-breeding site by country was tested with a binomial model (sub-Saharan site yes or no ~ country) and difference in the number of sub-Saharan non-breeding sites by country was tested with a Poisson model. None of the migration periods to change non-breeding sites south of the Sahara occurred within one week of the equinox.

Because we found use of more than one non-breeding site for some whinchats, and the difficulty of determining whether these were additional non-breeding sites, or the first stopover of migration – particularly when these additional sites are located approximately on the migration route and used just before the spring migration – we carried out the analyses for the predictions below with and without additional non-breeding sites, where appropriate. A non-breeding site was defined as simply any sub-Saharan (less than 18 degrees of latitude) stationary period during the non-breeding period. Sixteen birds had more than one potential non-breeding location, involving 19 additional non-breeding sites and migratory legs. For tests below involving migration leg daily travel speed, migration stopover duration and migration duration we focus on the period when a whinchat departed to cross the Sahara in the spring and remove sub-Saharan stationary periods at the start of analysis.

**Prediction 3.** We predicted that Liberian birds would have longer total migration distances, use more stopover sites and not show a loop migration pattern. Total migration distance was calculated as the sum of all distances between every stationary period. Total stopovers were simply the total number of stationary periods on migration between the non-breeding and breeding site. This analysis was unaffected by any imprecision in latitude that might arise at the equinox because any over or underestimates in latitude for a stopover period, would cancel out and so broadly not affect the overall total migration distance between the main non-breeding site in Africa and the breeding site in Europe. Loop

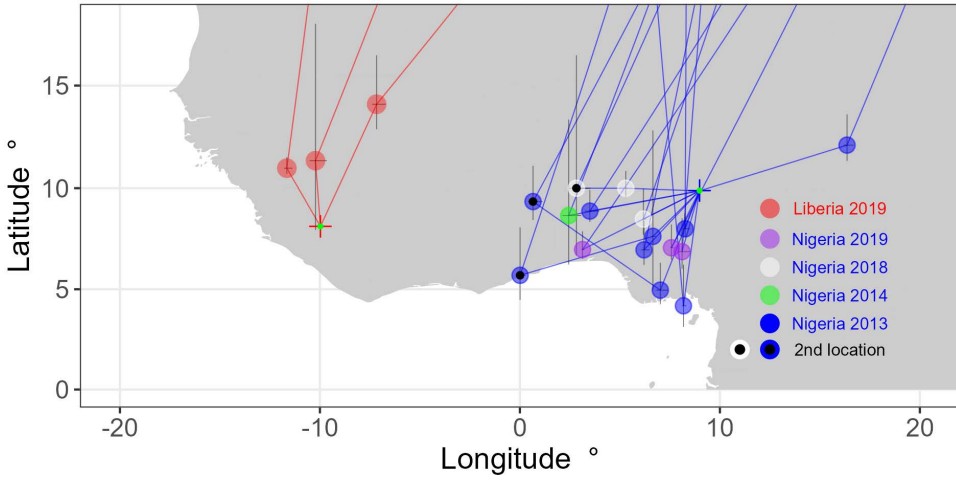

**Fig 3. The three (19%) Liberian birds and 13 (27%) Nigerian birds that used sub-Saharan sites away from their main non-breeding location (where they were tagged and retrapped a year later).** Three Nigerian birds (6%) used two sites before crossing the Sahara. Each plotted point is the location of an additional non-breeding site, with filled points indicating a further second location (i.e., three non-breeding locations in total for the individual). Locations are plotted with one standard error of estimated latitude and longitude assuming the same sun elevation angle as measured on the main non-breeding site.

migration was tested by assessing how longitude varied between spring and autumn migration: a significant difference in mean longitude by season would indicate a loop migration, and a model including the interaction between season and country then tested whether any degree of loop migration (difference in longitude) varied by country.

**Prediction 4.** We predicted that overall migration duration and phenology would be different between birds tracked from Nigeria and Liberia, but these differences would be explained by breeding latitude or overall migration distance. Overall migration duration was the total number of days between departure from the main non-breeding (or the last site used in sub-Saharan Africa, <18 degrees latitude for those birds that used more than one non-breeding site) and arrival at the breeding site, or vice versa, and was log-transformed to meet model assumptions. Phenology models considered the Julian date of departure from the non-breeding site in the spring, the duration of spring migration (the number of days between departure from sub-Saharan Africa and arrival at the breeding site), the duration of the breeding stationary period in days, the Julian date of departure from the breeding site in autumn or the Julian date of arrival at the non-breeding site.

**Prediction 5.** We predicted that migration leg distance and migration duration of a leg (and therefore speed of daily migration during a migration leg), and total stopover duration, would be similar. Migration leg distance was the distance between sequential stopover/stationary sites; migration leg duration was the number of days this leg took; migration leg speed was simply leg distance/duration; and stopover duration was the number of days spent between migration legs. Migration leg travel speed and stopover duration were log-transformed in all models to meet model assumptions. Analyses were also run excluding the 16 migration and stopover periods that occurred within one week of the equinox to determine how imprecision in calculation of latitude during these periods affected the results.

Raw data for distances, durations, phenology and breeding location are summarised in S2 Table.

Mean values are presented with one standard error (se) and $R^2$ values were adjusted in all cases. Predicted values were calculated from models using the predict function in the MuMIn library in R [54] and using mean values for all other variables in the model; marginal variance in mixed models was calculated using the r.squared GLMM function in the MuMIn library.

The study was carried out in Nigeria and Liberia where no licences are required for the procedures used. Nevertheless, this study was carried out under the ethical guidelines stipulated by the AP Leventis Ornithological Research Institute Scientific Committee (APLORI is the only ornithological research institute in Nigeria) based on the Association for the Study of Animal Behaviour guidelines, the University of St Andrews School of Biology's Ethical Committee and those of the British Trust for Ornithology's ringing scheme. All personnel involved in fieldwork – either catching, colour-ringing or tagging birds had BTO ringing licences. MB, WC and RP have been licensed to fit geolocators in the UK or the EU in other studies.

## Results

### 1. Breeding range, overlap and connectivity

Liberian whinchats migrated along the western side of West Africa and Europe, whereas Nigerian birds migrated through the eastern side of the Mediterranean and Europe; both in a similar north-northeast direction (Fig 1). Breeding sites were largely in Scandinavia and north-western Russia for Liberian birds, although one bird bred in Switzerland (range 2.24 million km$^2$). In contrast Nigerian birds bred in Eastern Europe and south-west Russia (range 2.11 million km$^2$). There was some overlap in ranges: 19% of the Liberian breeding range overlapped with the Nigerian breeding range; 27% of the Nigerian breeding range overlapped with the Liberian range; 11% of the total breeding range of both origin countries overlapped (Fig 2). Mean migratory spread for Liberian birds was 1009 km (159–2827 95% CI); mean migratory spread for Nigerian birds was 658 km (124–1687 95% CI).

### 2. Multiple non-breeding site use

Three out of 16 (18.8%) Liberian birds and 13 out of 48 (27.1%) Nigerian birds used a second sub-Saharan non-breeding site on average 503.6 km±47.0 SE away from the main non-breeding site, for a mean duration of 28.5±3.0 SE days; a

further 3 Nigerian birds also used a third sub-Saharan non-breeding site, with the third site being on average 630 km ± 181 SE away from the second site, for a mean duration of 36.3 ± 18.3 days. The three Liberian birds all used subsequent sites to the north of the main non-breeding site, and so shortening the distance to the breeding ground, but this was only the case for one Nigerian bird (Fig 3). Instead, 12 out of the 13 Nigerian birds moved west or south, with the majority going south-west: four birds moved over 500 km further south. The overall total maximum distance of movement recorded was approximately 1,442 km (note the latitudinal part of this will be subject to location error) by a bird that moved south approximately to the coast of Nigeria and then approximately to north-west to Ghana, spending 35 days and 17 days at the two sites respectively; a second bird also approximately reached Ghana, travelling southwest, 356 km and 765 km in two legs, with a stopover of 2 and 19 days respectively. The subsequent flight for this bird crossed from approximately the southern coast of Ghana to approximately the Mediterranean coast of Tunisia in a single leg of around 3,538 km over a 5-day period. The total distance travelled between the main Nigerian non-breeding site and the Mediterranean coast for this bird was therefore approximately 4,659 km, approximately 2,130 km further than the direct, great circle distance from its main non-breeding site in Nigeria to where it eventually arrived on the Mediterranean coast. One final bird also used multiple sites. The bird was tagged in 2013 was only recaptured in 2014, 18 months after tagging. This bird spent the entire non-breeding period of 2013−14, at least until March 25th 2014 when the logger battery ran out, at a site about 720 km to the west approximately on the border of Chad and Cameroon.

The probability of having more than one sub-Saharan non-breeding site was not significantly different by country (z = 0.7, P = 0.51, N = 64 birds) and the number of sub-Saharan non-breeding sites was not significantly different by country (z = 0.9, P = 0.36, N = 64 birds).

### 3. Distances, number of stopovers and loop migration

Total migration distance was marginally different for spring and autumn migration (t = 1.8, P = 0.078), but significantly shorter for Nigeria (1001 ± 199 km shorter, t = −5.0, P < 0.0001), and there was no significant change in the difference between spring and autumn across countries (t = 1.0, P = 0.29; marginal $R^2$ = 0.29; N = 92 distances from 51 birds; Fig 4). Excluding sub-Saharan stopovers prior to spring migration gave similar results but increased the difference in spring and autumn to give a significant difference (Table 1): total migration distance was significantly longer for autumn versus spring (284 ± 83 km longer, t = 3.4, P = 0.0015), and significantly shorter for Nigeria (1032 ± 189 km shorter, t = −5.4, P < 0.0001), and there was no significant change in the difference between spring and autumn across countries (t = 1.5, P = 0.13); marginal $R^2$ = 0.34; N = 92 distances from 51 birds.

There were significantly fewer stopovers in spring for Nigerian birds compared to Liberian birds (−1.0 ± 0.26, t = −3.8, P, 0.0004), but a similar number of stopovers for Liberian birds in autumn compared to spring (−0.3 ± 0.3, t = −0.9, P = 0.38), and a marginally significant difference in the change in the difference between spring and autumn across countries, with birds tagged in Nigeria increasing the number of stopovers in autumn, while they decreased for birds tagged in Liberia (change in difference, 0.72 ± 0.39, t = 1.8, P = 0.075; N = 92 complete spring or autumn passage periods, N = 51 birds; marginal $R^2$ = 0.16; Fig 5). Excluding sub-Saharan stopovers prior to spring migration gave a similar result but increased the statistical significance. There were significantly fewer stopovers in spring for Nigerian birds compared Liberian birds (−1.1 ± 0.25, t = −4.4, P < 0.0001), but there were a similar number of stopovers for Liberian birds in autumn compared to in spring (−0.2 ± 0.3, t = −0.5, P = 0.64), and a significant difference in the change in the difference between spring and autumn across countries, with Nigeria increasing the number of stopovers in autumn, while Liberia decreased them (change in difference, 0.83 ± 0.37, t = 2.2, P = 0.032; N = 92 complete spring or autumn passage periods, N = 51 birds; marginal $R^2$ = 0.25; Table 1).

The difference in mean longitude of spring and autumn stopovers across individuals shows the degree of loop migration, with no difference if birds take a similar route on spring and autumn migration. The mean longitude of spring and autumn stopovers was significantly different by country (Nigeria −5.7 ± 2.6 degrees, t = −2.2, P = 0.031; N = 41 birds, one

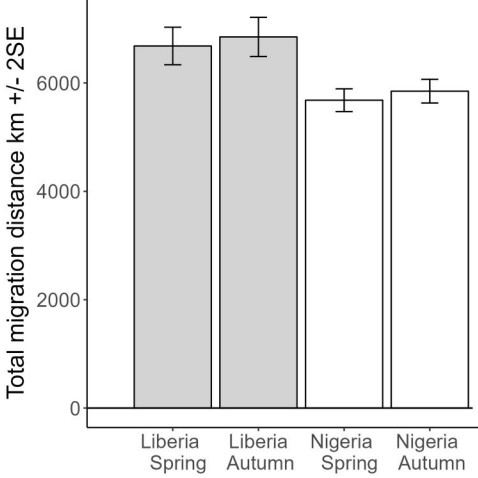

**Fig 4. Predicted values from a mixed model of total migration distance (sum of all great circle distances between stationary periods for spring and autumn respectively).** With country and season as main effects, including individual bird tracked as a random effect, marginal $R^2 = 0.29$, N = 51 birds and 92 complete spring or autumn migrations. The interaction term country*season was not significant.

**Table 1. Summary statistics for migration characteristics of 64 Liberian and Nigerian whinchats.**

| | | Spring | | Autumn | | Spring | | Autumn | |
|---|---|---|---|---|---|---|---|---|---|
| | | Mean | SE | Mean | SE | Mean | SE | Mean | SE |
| | | | | | | | | | |
| Total migration distance | km | 6682 | 173 | 6849 | 180 | 5681 | 105 | 5848 | 109 |
| Total migration distance* | km | 6580 | 164 | 6864 | 171 | 5548 | 99 | 5832 | 103 |
| Total number of stopovers | | 3.5 | 0.2 | 3.2 | 0.3 | 2.3 | 0.1 | 3.0 | 0.1 |
| Total number of stopovers* | | 3.4 | 0.2 | 3.2 | 0.3 | 2.3 | 0.1 | 3.0 | 0.1 |
| Mean migration leg distance** | km | 2043 | 67 | | | | | | |
| Mean migration leg distance* | km | 2008 | 156 | 2047 | 214 | 2411 | 106 | 1962 | 109 |
| Mean migration leg duration** | days | 3.1 | 0.1 | | | | | | |
| Mean migration leg duration* | days | 3.1 | 0.1 | | | | | | |
| Mean leg travel speed* | km/h | 26.7 | 2.6 | 35.6 | 4.7 | 31.9 | 2.1 | 25.5 | 1.8 |
| Mean stopover duration | days | 5 | 1.2 | 7.1 | 1.9 | 7.5 | 1.8 | 10.5 | 2.7 |
| [1]Mean stopover duration* | days | 5.1 | 1.3 | 6 | 1.7 | 5.4 | 1.2 | 11.0 | 2.7 |
| [1]Total migration duration* | days | 25.8 | 2.6 | 23.8 | 3 | 18.6 | 1 | 37.1 | 2.2 |
| Mean travel speed during migration period* | km/h | 8.9 | | 9.6 | | 9.6 | | 5.1 | |
| [2]Mean hours of migration leg day stationary if travelling at 43 km/h when flying | | 8.9 | | 8.6 | | 5.9 | | 9.3 | |

*Excluding sub-Saharan movements or stationary periods prior to spring crossing of the Sahara

**Null model was the best fit for the data

[1]These two values should be added together to give a complete average migration duration

[2]Calculated as (observed leg duration minus predicted duration for 43 km/h)/observed duration

mean difference per bird, adj. $R^2 = 0.09$). Liberian birds showed no significant difference in the longitude of stopovers spring compared to autumn migration ($2.8 \pm 2.8$ degrees, t = 1.0, p = 0.33; N = 40 stopovers from 4 birds; marginal $R^2 = 0.02$), whereas Nigerian birds showed significantly more easterly stopovers in autumn indicating a consistent loop migration on average ($7.8 \pm 1.3$ degrees, t = 6.2, P < 0.001; N = 119 stopovers from 33 birds; marginal $R^2 = 0.23$). In an overall model,

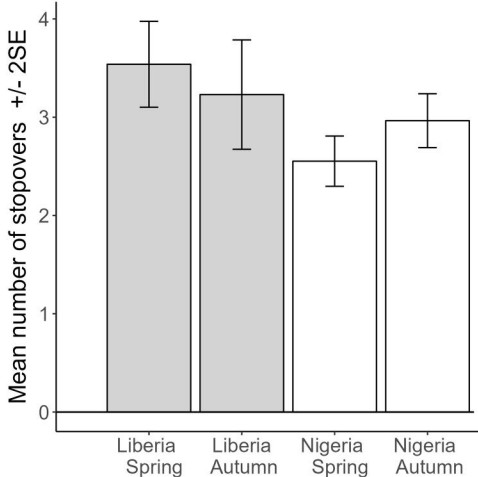

**Fig 5. Predicted values from a mixed model of number of stopovers, with country and season as main effects, including individual bird tracked as a random effect, and testing the interaction of country\*season.** Marginal R$^2$=0.16, N=51 birds and 92 complete spring or autumn migrations. The interaction term is marginally significant, with the main significant difference being between Liberia and Nigeria spring birds.

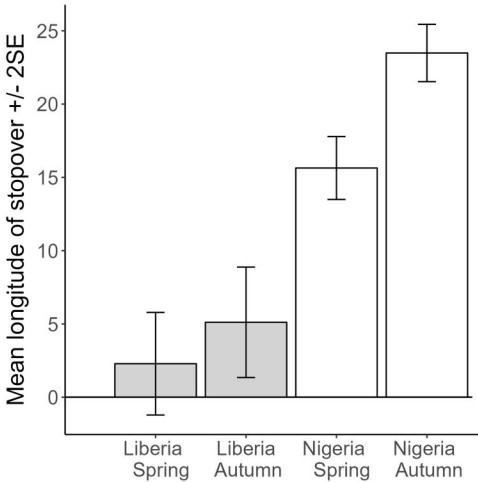

**Fig 6. Predicted values from a mixed model of longitude of stopover, with country and season as main effects, including individual bird tracked as a random effect, and testing the interaction of country\*season.** Marginal R$^2$=0.52, N=41 birds and 159 stopovers. The interaction term is marginally significant: a larger increase in longitude for autumn vs spring for Nigerian birds, compared to Liberian birds.

the interaction was marginally significant with a greater increase in longitude for autumn vs spring of 5.0±2.7 degrees for Nigerian birds, compared to the increase for Liberian birds (2.8±2.3 degrees, t=1.8, P=0.06; N=159 stopovers from 41 birds; Fig 6).

## 4. Overall migration duration and phenology

Overall migration durations (starting from when a whinchat started to cross the Sahara in the spring and finishing when the whinchat reached it main breeding site, or when the whinchat left its main breeding site and returned to the main non-breeding site where it was tagged) were shortest for birds tagged in Nigeria migrating in Spring and longest for

Nigeria birds migrating in Autumn (35% longer than the average duration of migration for birds tagged in Liberia, and Nigeria in spring). Migration duration for spring and autumn migration in Liberia was similar (t = −0.5, P = 0.59), but comparing Liberia to Nigeria, migration duration in spring was significantly shorter for Nigeria (t = −2.6, P = 0.012), but then significantly increased in duration in autumn (t = 4.6, P = 0.0001), controlling for the significant increase in migration duration with breeding latitude (t = 2.0, P = 0.048; N = 91 spring or autumn migrations from 51 birds; marginal $R^2$ = 0.49; Fig 7). Migration durations were very short for several birds. One whinchat moved from its sub-Saharan non-breeding site to its northern European breeding site in 7 days, and 8% of complete migration durations were 14 days or less (N = 89).

Nigerian birds crossed the Sahara (left the non-breeding grounds for their spring migration) significantly later than Liberian birds (8.3 ± 3.1 days later than the 9th April mean departure date for Liberia, t = 2.7, P = 0.010, $R^2$ = 0.10, N = 51 birds, controlling for the non-significant effect of breeding latitude, set at mean value of 55.7 degrees for these predicted values; 4.8 ± 2.1 days later when not controlling for latitude, t = 2.3, P = 0.023). If we control for distance travelled, instead of latitude (these are correlated) during spring migration, we found a similar effect (7.2 ± 3.0 days later, t = 2.4, P = 0.021, $R^2$ = 0.07, N = 51 birds). Birds breeding at higher latitudes arrived at the breeding site later (0.74 days ± 0.2 SE for every degree of latitude further north, t = 3.8, P = 0.0004) independently of non-breeding country (t = 0.5, P = 0.63): $R^2$ = 0.20, N = 51 birds that made complete spring migrations. The duration of the stay at the breeding site was similar between the two countries: Liberia 111 ± 5 days and Nigeria 107 ± 3 days (difference −11.8 days ± 7.7 t = − 1.5, P = 0.13, with a marginally significant effect of latitude, 1.0 days ± 0.6 SE for every degree of latitude further north, t = −1.8, P = 0.074; $R^2$ = 0.04, N = 42 birds). Departure date from the breeding grounds for the autumn migration was also similar between the two countries, with the mean being the 23rd August (± 2.6 days): difference between the countries, Nigeria −7.6 ± 8.1 days, t = −0.9, P = 0.35, with a nonsignificant effect of breeding latitude, t = −0.4, P = 0.66, $R^2$ = 0.001, N = 42 birds. Arrival at the wintering grounds after autumn migration was also similar between the two countries, with mean arrival date being 30th September (± 2 days): difference between the countries Nigeria plus 7.7 ± 5.8 days, t = 1.3, P = 0.19, with a nonsignificant effect of breeding latitude, t = 0.3, P = 0.74, $R^2$ = 0.001, N = 40 birds.

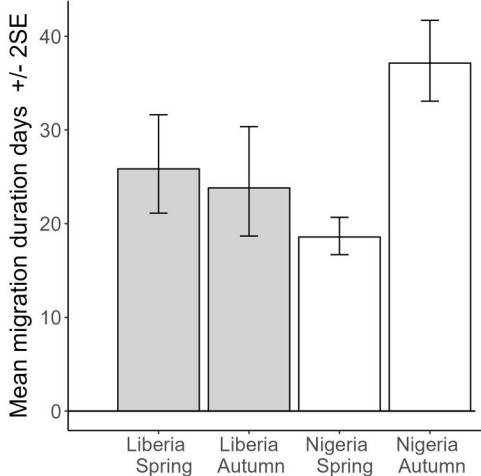

**Fig 7. Predicted values from a mixed model of overall migration duration, with country and season as main effects, controlling for breeding latitude (predicted values set to mean breeding latitude of 55.7 degrees), including individual bird tracked as a random effect, and testing the interaction of country*season.** Marginal $R^2$ = 0.15, N = 51 birds and 92 complete distances. All bars are significantly different apart from Liberia spring and autumn. Note the data here starts migration from the last sub-Saharan non-breeding site where multiple non-breeding sites were used and finishes when the bird returns to the main non-breeding site, where the bird was tagged.

## 5. Migration leg distance, duration and travel speed, and stopover duration

There was no significant variation in the distance migrated in a leg by country or season, or their interaction: on average whinchats flew 2,043±67 km (N=298 distances from 64 individuals) per migration leg (this was 2,053±69 km excluding the 16 equinox samples). Excluding sub-Saharan stopovers prior to spring migration, the interaction model best explained migration leg distance (ΔAIC$_c$=−0.5 compared to the main effects model including country and season and ΔAIC$_c$=−3.8 compared to the null model), with spring legs being longer for Nigerian birds than Liberian birds (402.1 km±190SE, t=2.1, P=0.036), no difference between season for Liberian birds (t=0.1, P=0.88) but a trend for a shorter migration leg length for Nigerian birds in autumn (−488 km±309SE, t=−1.5, P=0.11; Table 1).

There was no significant variation in the duration of a migration leg by country or season, or their interaction: a migration leg lasted on average 3.1±0.1 days (N=298 durations from 64 birds; the null model was best supported: ΔAIC=−3.1 compared to the main effects model including country and season). Excluding the 16 equinox samples (3.1±0.1 days) or excluding sub-Saharan stopovers prior to spring migration gave very similar results (Table 1).

Excluding sub-Saharan stopovers, there was, however, significant variation in the speed of travel during a migration leg by country and season: speed was marginally significantly faster in spring than autumn for Liberia (t=1.8, P=0.070), similar in spring for Liberia and Nigeria (t=1.5, P=0.11), but Nigeria showed a significantly slower speed during autumn (t=−2.8, P=0.0059; but note the model was a very poor fit to the data, marginal R$^2$=0.03, N=279 leg speed estimates from 64 birds; Fig 8). Overall, an average speed of travel (note this is not flying speed because it also includes stationary periods of less than 2 days) during a migration leg of 2043 km/ (3.1 days x 24 hours) = 27.5 km/h was therefore reasonably representative across populations and seasons.

Excluding sub-Saharan stopovers prior to spring migration, some of which were likely to be additional non-breeding sites, stopover durations during spring migrations were similar across countries (t=0.5, P=0.61) and autumn migration stopover durations for Liberia were similar to spring (t=0.9, P=0.37), but autumn migrations stopovers for Nigerian birds were nearly double the other classes (t=2.7, P=0.0075; marginal R$^2$=0.25; N=186 observations from 64 birds; Fig 9; excluding the 16 equinox samples gave almost identical results). When we include the generally longer sub-Saharan "stopover" periods prior (or at the start) of spring migration, then the pattern changes considerably. Average stopover duration was now significantly longer for Nigerian birds (t=3.3, P=0.0013) and for autumn migrations generally (t=3.2,

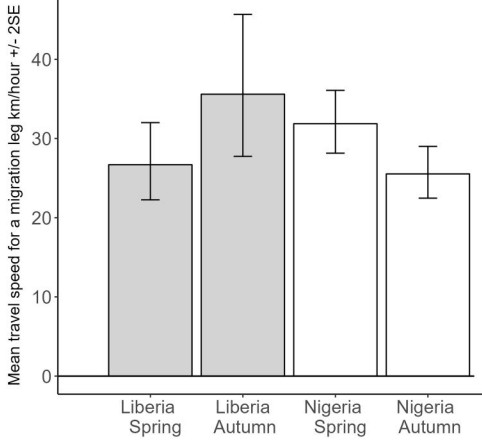

**Fig 8. Predicted values from a mixed model of speed of travel during a migratory leg (excluding sub-Saharan movements), with country and season as main effects, including individual bird tracked as a random effect, and testing the interaction of country*season.** Marginal R$^2$=0.03, N=64 birds and 279 complete migration legs. The interaction term is significant with main difference being the greater decrease in speed between spring and autumn migrations for Nigeria birds.

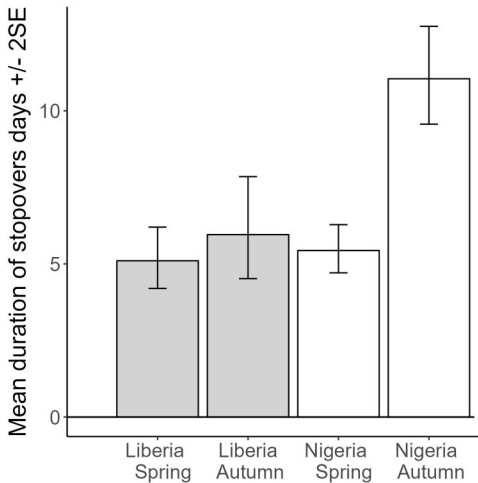

**Fig 9. Predicted values from a mixed model of stopover duration, with country and season as main effects, including individual bird tracked as a random effect, and testing the interaction of country\*season.** Marginal $R^2 = 0.24$, N = 64 birds and 186 stopovers. Only the autumn Nigeria bar is significantly different. Note the data here excludes all stationary periods south of the Sahara in spring.

P = 0.0018) but there was no interaction between country and season (t = 1.4, P = 0.17; marginal $R^2 = 0.11$; N = 205 stopovers from 64 birds; Table 1; excluding the 16 equinox samples gave almost identical results).

## Discussion

We tracked 64 whinchats over their annual migration cycle from non-breeding sites in Nigeria and Liberia, to Europe and back, across multiple years. We showed that both groups had similarly large estimated breeding ranges, according to land availability to the north of their non-breeding locations. We showed large predicted migratory spread for both populations and so some overlap of breeding ranges. We showed a similar proportion of the two populations using more than one non-breeding site, but with greater use of secondary non-breeding areas to the south for Nigerian birds according to greater land-availability to the south of this population. We showed predicted differences in distance and duration of migration, according to the availability of non-breeding areas to the north, and both populations showed similar migration distances in spring and autumn, with a biologically small increase in autumn migration distance; Nigerian birds showed longer stopover duration in autumn relative to spring and Liberia, vice-versa. We showed loop migration only in the Nigerian population, again as predicted according to land availability. We showed that predicted migration leg duration and distance were similar for both populations (but with some minor consistent variation in speed), and stopover duration according to selection for general energy minimisation. Although excluding sub-Saharan stopovers, Nigerian legs were probably longer distance than Liberian legs in spring. Finally, we showed that overall migration durations and phenology were similar as broadly predicted, reflecting the longer distances and higher latitudes for the breeding areas of Liberian birds, apart from a longer autumn migration for Nigerian birds, possibly reflecting their loop migration. Overall, the differences we identified seemed to support hypotheses that geography, and possibly stopover site quality, creates differences in some migration characteristics, but many characteristics remain similar across populations because of overarching selection for time and energy minimisation during migration.

### 1. Breeding range, overlap and connectivity

The geographical range in breeding areas was as expected from the differences in geographic location between Liberia, in the southwest and Nigeria to the west of West Africa. There are essentially no breeding areas to the northwest of

where we sampled in Liberia, and few potential other non-breeding areas to the south. This apparently trivial result (birds can only occur where there is suitable habitat) is important however because many migratory connectivity analyses fail to consider this [20]: that patterns of connectivity are actually better explained by geography than by functional ecology of migrants [8]. Migrants move from their breeding grounds to the closest available non-breeding site that minimises migration costs relative to gains at the non-breeding site [24,55]. But this is with the caveat that major shifts in the availability of breeding areas due to long term climatic change in Europe [56], coupled with very strong selection for natal site philopatry (i.e., returning to the same breeding area), can lead to non-optimal routes [57].

Migratory spread was high, although higher for Liberian whinchats than shown previously using a smaller sample size from Nigeria [40], and similar to whinchats tagged in the UK [38]. Values of migratory spread for all whinchat populations analysed (now N = 3) fall within the range shown by 90% of 86 landbird species analysed, and reasonably close to the median value [20], suggesting that whinchats have "typical" low connectivity. Nevertheless, the distance between the starting locations in Liberia and Nigeria inevitably means that the two breeding populations will have comparatively low overlap [i.e., high connectivity if simply describing the observed pattern rather than the underlying evolutionary processes of dispersal that drive connectivity, see 20]. The actual range of the populations and degree of overlap is however a consequence of sample size, making further conclusions unreliable. As shown in Fig 2, a single location for a bird that bred in Switzerland greatly increases the breeding range for the Liberian population; a further likely breeding point in northern France had to be excluded because we could not tell whether the tag failed on route, rather than just as the bird reached its breeding ground, and this location would also have a very large effect on the observed range. Similarly, the lack of birds from Liberia breeding in Western European countries such as the France and UK is very likely simply to represent the small sample size of tagged birds from Liberia, and the disproportionately small western European populations compared to eastern and northern areas [37,58]. Whinchats tagged in the UK, sampling this relatively small population directly, for example, utilised a non-breeding site less than 175 km from the Liberian population in the present study, and ranged right across West Africa from Sierra Leone to Nigeria [38].

## 2. Multiple non-breeding site use

A small proportion of whinchats used more than one non-breeding site (about 20% of birds in both populations) and this is very similar to the only other study which has tagged whinchats [20%, N = 20 tagged birds from 3 separate UK areas, 38]. Use of a main non-breeding site, followed by additional non-breeding site, particularly as the spring migration approaches has been recorded in a number of species now [see discussion in 3, 38 for specific examples, 59]. Use of multiple sites during the non-breeding period, particularly movements that increase migration time and distance [where survival is relatively low 13] must either provide compensatory benefits, such as better resources [10] or arise because of a reduction in quality of the first non-breeding site below a critical threshold [60]. This potentially important behaviour – in the sense that it indicates the ability to compensate for a change in quality of non-breeding habitat, and the related issue of the scale at which juveniles select their first (and usually only) non-breeding site, is critically understudied [9].

Although specific distances moved between non-breeding sites in terms of latitude are likely to be inaccurate, our results show large latitude changes and unequivocal longitude changes, and so probably robustly highlight a number of interesting points about multiple non-breeding sites. Some non-breeding sites can perhaps be viewed as clearly adaptive longer term staging sites (stopovers), such as the Liberian birds that moved closer to the Sahara edge at the start of the spring so minimising the width of the barrier to be crossed by a single flight. But many second or even third non-breeding sites may greatly increase overall migration distance in the spring, as well as adding significant migration costs during the non-breeding season, as shown by the Nigerian individual that increased its total first migration flight distance to North Africa, crossing the Sahara, by 688 km (a 24% increase compared to the distance if the bird had left from its first non-breeding site, as the majority of whinchats did). This suggests that intra-African migrations (i.e., between non-breeding sites south of the Sahara) to find new sites in a bird's first non-breeding season, or to revisit known sites in

subsequent non-breeding seasons, may be relatively low cost. Any costs must be offset by benefits. In this study, the bird that increased it first migration flight by 24%, likely crossed the entire width of West Africa in a single leg (a flight without more than a few hours stopover per day) lasting 5 days suggesting that fattening rates at a site in southern Ghana where the wet season will have commenced, rather than end of the dry season in central Nigeria, were much more favourable. This hypothesis is supported by apparent survival rates being similar between whinchats that use multiple sites and those that only used a single site in Nigeria, suggesting that the strategies can be equally good [41].But it is important to note that we only have movement data from surviving birds.

### 3. Distances, number of stopovers and loop migration

Liberian migration distances were longer than Nigerian, reflecting the availability of land, and the corresponding population sizes that were sampled, as mentioned above. Because of low connectivity, Liberian birds will originate from all over Europe, but only where there is land, therefore we were more likely to sample birds that had arrived from the far north and west. In Nigeria, whinchats can arrive directly from breeding areas from many directions so, on average, migration distances of a sample will be lower. This is another apparently trivial result but is important because it shows how the variable geography of Europe might lead to different characteristics of non-breeding populations for low connectivity species. For example, Liberian populations had a greater proportion of birds making longer migrations, and so are exposed to longer periods when survival is expected to be lower [13]. These longer migrations result in Liberian populations having a greater proportion of birds making more stopovers and so increasing "multiple jeopardy" [7]. The reverse will not apply, however, for any species of migrant passerine that migrates on a broad front, sampled on the breeding ground, because Africa is more or less equally available for any breeding population.

The migration route of Nigerian birds was further east in autumn than during spring (a loop migration), but for Liberian birds, deviations from the more direct spring route were significantly less and in either direction: there was a small increase (less than 5%) in autumn migration distances reflecting this. The only other whinchat tracking study, of UK breeding birds, also showed a tendency for a direct more easterly desert crossing in the spring and a westerly (but similarly longer) route in autumn, suggesting an anti-clockwise loop migration [38]. It is possible that the lack of a significant difference in both studies is a consequence of sample size, and loop migrations occur in westerly whinchats as they do for Nigerian whinchats. Westerly autumn migration around the edge of the Sahara has also been shown for other migrant passerines [e.g., 61, 62]. Such loop migrations are hypothesised to be detours in response to avoiding headwinds that dominate central areas of the Sahara in autumn [e.g., 63] and although they may result in longer migrations, they may be optimal energetically [e.g., 64].

### 4. Overall migration duration and phenology

Overall migration duration varied according to distance and latitude, apart from a longer autumn migration for Nigerian birds, reflecting their longer stopover duration as discussed below. There was a later spring departure date for Nigerian birds, but this did not reflect their shorter migrations, or the higher breeding latitude of Liberian birds. When we controlled for either breeding latitude or distance, we still found a predicted difference in departure date of 8.3 and 7.2 days respectively, showing that for any similar latitude or distance, Nigerian birds left on average later than Liberian birds. But this then leads to the question, why do Liberian birds not adjust by starting their migration later than Nigerian birds? Survival should be maximised by staying on the non-breeding ground, where we know survival is very high [15], and foraging conditions for insectivores radically improve with the onset of rains in April. Also, survival should be maximised by minimising the time spent migrating. However, the mean travel speed of all whinchats during a migratory leg was similar and also overall when migrating (migratory legs and stopovers), apart from the slower overall rate of travel for autumn Nigeria birds. This suggests that perhaps Liberian whinchats were time minimising, but in both spring and autumn, and slower travel rates for autumn Nigeria birds reflect constraint – either poor migrating conditions such as headwinds, or poor stopover sites.

Departure times from West Africa may also be determined by local conditions rather than timing of breeding [65] so earlier departure by Liberian birds may reflect more favourable conditions there (savannah in a humid forest mosaic rather than degraded, near Sahelian savannah with much grazing and farming in Nigeria).

The major difference that led to timing differences between spring Liberia and Nigeria birds was the number of stopovers – Liberia birds had 43% more stopovers than Nigerian birds, whereas the increase in migration length was only 18%. Liberian birds were probably therefore spending more time at stopovers than they needed to refuel. In spring, adjusting migration arrival time through increasing stopovers may allow better adjustment of arrival times [66], particularly for Arctic breeding birds that might pay a high survival cost for arriving early. Timings will have been complicated by geographic location of breeding areas, with some whinchats breeding well above the Arctic Circle, and so these areas will only suitable for whinchats later in the season even up to the end of May [37]. The range in breeding site latitude across all whinchats in this study was 26.2 degrees, which gives a predicted difference in arrival dates of 19.3 days (using the model in section 5 from this study). Arrival date was best predicted by latitude, rather than non-breeding country in accordance with this, and this along with the country difference in departure date controlling for latitude, suggests that Liberian birds in particular adjust their migration speed to phenology during spring migration, rather than through departure date. Long distance migrant birds, such as pied flycatchers *Ficedula hypoleuca* use conditions at the non-breeding area, en-route and local to the breeding area, to time migration [65] and the relative importance of these conditions varies across populations [67]. It has been shown that many long distance migrant populations will have to add an additional stopover due to future climate change [68]. This type of adjustment is likely already commonplace for whinchats and other migrants, because the serial residency hypothesis and corresponding low connectivity means that individuals from the same breeding population have varied starting points, routes and phenology to deal with every year [9], promoting traits that allow flexibility and generalism that are correlated with adaptation to phenological change [69,70].

## 5. Migration leg distance and duration, and stopover duration

Migration leg duration and distance was similar for both populations, with an average of 2,043 km travelled in 74.4 hours with an average travel speed of 27.4 km/h. The migration leg characteristics in spring did not vary by population suggesting that the values represent an average migration capability. Typical flight speed of migrating birds measured with radar is between 40–55 km/h with chats at the lower end of this [71]; flight speed of Cyprus wheatears *Oenanthe cypriaca*, a species of similar size and morphology to whinchats, in their likely single non-stop flight of 2,500 km from Cyprus to Eritrea over the Sahara is 43.1 km/h [72]. If we then assume a similar flight speed for whinchats, then 2,040 km can be covered in 47.3 hours, suggesting that 27.1 hours of the average migration leg time was spent stationary. Because of variable average migration leg distances (also reflected in variation in migration leg speed) per day, this would equate to a minimum of 5.9 (Nigeria spring) to a maximum of 9.4 hours (Nigeria autumn) of a migration leg in the daytime that a whinchat could be on the ground foraging or resting (Table 1). It is important to note that although there may be significant individual variation and that we assume great circle shortest distances between stopovers, the key result is that at least half of all daylight hours are available to a whinchat to forage and rest even during migration legs.

Many small passerines, start migration just after sunset [73], migrate predominantly overnight (although many extend their continuous flights into the following day [26]) and then rest or feed during the day [74]. Some small passerine species, however, make migration legs without any stops, even during the day [75,76], and it is clear that the occurrence of continuous migration night and day is common and varies between and within species, but is particularly associated with barrier crossing [77]. The results of our study suggest that whinchats make daily stops: an average migration leg of 74.4 hours is likely to be carried out over 2 nights and 3 days (assuming departures just after sunset), with potential stops of about 9.0 hours during the three days, and each nocturnal flight extended into the day by up to a few hours. This pattern then makes sense of the light data from many of the individuals in this study, where there may be a gradual change in sunset and sunrise over periods of more than a week (e.g., see Appendix Fig 1), particularly when not crossing barriers,

when whinchats must be using the approximately 75% of daylight when they are not flying for refuelling, so sustaining a continuous migration without the need for longer stopovers. In short, daily stopovers are important for migrating whinchats for foraging and resting, and even with barriers to cross, this may represent the typical migration pattern of whinchats. Most whinchats left their Nigerian primary non-breeding area in the spring with lower fuel reserves than necessary to cross the Sahara in a continuous flight [78], suggesting that daily refuelling – even in the Sahara and Mediterranean sea region – may be a typical part of whinchat migration. This seems to make sense because whinchats can utilise any open habitat with sparse vegetation and are extreme generalist foragers in both Africa [42,60] and Europe (Cramp 1988), and so are probably able to utilise the marginal habitats and limited vegetation available in many parts of the Sahara [79], and the numerous islands of the Mediterranean [80] when necessary. Nevertheless, crossing the Sahara results in faster migration, longer legs and longer stopovers post crossing on average, suggesting that it is an area where the potential for nocturnal flight/daily foraging during migration is reduced for many whinchats [39].

Although migration leg characteristics were broadly the same across populations and seasons, we found that Nigerian birds had much longer stopovers in autumn. And if we consider migration legs only above or crossing the Sahara then Nigerian autumn migratory legs were about 20% (c. 400 km) longer. Shorter duration spring migrations are predicted by time-minimisation [81], in order to reach the breeding ground as quickly as possible to reduce risk associated with long stop-overs [82], to ensure arriving early to be able to time breeding to match peaks of food availability [83], and to gain or maintain the best territories [84]. Timing rather than routes primarily determines stop-over duration and so overall speed of migration [85–87] and adaptive shortening or even dropping of daily stop-overs may be a common strategy for faster migration [88]. This should also apply to Liberia but there was no difference in stopover durations between spring and autumn despite clear seasonal differences in other migratory characteristics, suggesting that habitat quality of refuelling sites may be higher, on average, on the western spring routes. Stop-over site quality rather than presence of barriers is likely to be the key determinant of migration speed [89–91].

## Conclusions

Overall, the differences we identified seemed to support hypotheses that geography creates differences in some migration characteristics, but many characteristics remain similar across populations because of overarching selection for time and energy minimisation during migration. This study has a limited comparison of just two non-breeding areas and cannot realistically use individual variation as a guide to adaptation and constraints (because any one latitudinal location from a geolocator is unreliable), nevertheless, the general comparison of mean migratory characteristics from the two populations provides useful information.

Whinchats made successful migrations across a wide range of routes and distances, breeding areas, and times of year. Large variation in spring phenology, migration leg distances, stopover number and duration and overall migration duration demonstrate an overall capacity within the species to deal with climate change and change in habitat availability, indeed as they must have done on a much greater geographic and temperature change scale, particularly over the last 125,000 years to 10,000 years ago [56]. In particular, the pace of migration shown in this study is one that allows much time for foraging on route. For example, the fastest migrating whinchat in this study travelled from a sub-Saharan stopover site in the southern Central African Republic to its breeding site near Moscow (5054 km) in just 7 days, without any major stopover, travelling at 30.1 km/h on average, and so giving at least 7 hours per day not travelling (approximately 60% of daylight hours assuming an average 12 hour day, typical of migration period) and assuming a flying travel speed of 43 km/h. Although the lack of a major stopover is unusual, the speed of travel while migrating was typical. It takes about 25 g of extra body reserves, or twice the amount carried by a whinchat outside of the migration periods, to cover approximately 5000 km [see 78]. Assuming this whinchat left sub-Saharan Africa with a full fuel load, it would require a further 12 g of extra body mass, gained at an average rate per day of 1.7 g during the journey, and so a daily fuel deposition rate of 13% of lean mass: similar high values have been recorded in other migrant insectivorous passerines [92,93]. The whinchat

must have encountered good foraging conditions at each daily minor stopover site that allowed sufficient refuelling, including potentially, oases during its crossing of the Sahara.

Refuelling rate during migration (whether every day or during major stopovers) are therefore crucial to rapid migration and therefore reducing the time spent exposed to risk of mortality. The relationship between fuelling rate and site quality is clear [94], and this then highlights what is probably the biggest issue with respect to general migrant declines: stopover quality. Most migrants rely on stopover sites between their breeding and non-breeding sites over which they have relatively little control in the sense that their choice of sites is constrained by time and energy efficient routes. Non-breeding sites, and indeed breeding sites (although this is unlikely), can be poor and offer only low rates of refuelling, but migrants can simply spend longer fuelling before starting migration at these familiar, low risk sites (which can even be changed to better sites over the long non-breeding and breeding period if necessary). But every stopover site needs to provide safe refuelling conditions, within a very narrow time window (although perhaps this applies less to autumn migration which is under less time pressure, hence the result we observed for Nigerian whinchats). The probability that this occurs over all sites decreases with the number of sites and so many individual long-distance migrants will inevitably encounter a degraded stopover site, extending their migration and reducing their survival. Species like whinchats, that do well in anthropogenic habitats, may have relatively minimal decreases in survival when stopover sites change because of human population pressure, but others that require more specialist habitats may have large decreases in survival. But we have almost no information on this for migrant passerine species and once again the key to understanding migrant declines is to determine where mortality or carry over effects on survival and productivity operate most strongly over their entire annual cycle. For a long-distance migrant species like whinchat, we know key factors operating on the breeding ground [e.g., 95, 96] and the non-breeding ground [e.g., 15, 41], but we know nothing directly about the effect of stopover site quality, and its availability. Non-archival tags that can give information of where and when passerine migrants die [e.g., [97]], and particularly those on their first migration [e.g., 98] similar to the satellite tags available for larger birds are essential to solve this problem.

## Supporting information

**S1 Table: Comparison of biometrics between Nigerian and Liberian whinchats used in analyses.**
(DOCX)

**S2 Table: Raw data summary.**
(DOCX)

**S1 Fig. A visual illustration of the process of determining migration and stationary periods.**
(TIF)

## Acknowledgments

We particularly thank Judit Mateos, Ben Freeman, Alice Risely and Arin Izang for their fieldwork catching, tagging, relocating and recatching some of the whinchats. We thank Chris Hewson for provision of some of the tags as part of an earlier tag effect study.

## Author contributions

**Conceptualization:** Will Cresswell, Rob Patchett, Emma Blackburn.

**Data curation:** Will Cresswell.

**Formal analysis:** Will Cresswell.

**Funding acquisition:** Will Cresswell.

**Investigation:** Will Cresswell, Rob Patchett, Emma Blackburn, Malcolm Burgess.

**Methodology:** Will Cresswell, Rob Patchett, Emma Blackburn, Malcolm Burgess.

**Project administration:** Will Cresswell.

**Resources:** Will Cresswell.

**Supervision:** Will Cresswell.

**Validation:** Will Cresswell.

**Visualization:** Will Cresswell.

**Writing – original draft:** Will Cresswell.

**Writing – review & editing:** Will Cresswell, Rob Patchett, Emma Blackburn, Malcolm Burgess.

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
