## [Decision Letter · Decision Letter 0]

19 Mar 2025

PONE-D-25-07789Comparing migration of Whinchats Saxicola rubetra from Liberia and Nigeria: behaviours modified by differences in geography but unlikely to be constrained by themPLOS ONE

Dear Dr. Cresswell,

Thank you for submitting your manuscript to PLOS ONE. After careful consideration, we feel that it has merit but does not fully meet PLOS ONE’s publication criteria as it currently stands. Therefore, we invite you to submit a revised version of the manuscript that addresses the points raised during the review process.

Both reviewers agreed with the value of the dataset and were impressed with the sample size. They were, however, both unsure of some part of the methodology, in particular about handling of the geolocator data around the time of equinox and how you dealt with the uncertainty. Please clarify this, as well as responding to other suggestions made by the reviewers.

We look forward to receiving your revised manuscript.

Kind regards,

Shoko Sugasawa

Academic Editor

PLOS ONE

Journal Requirements:

2**.** Thank you for stating the following financial disclosure:

“Chris Goodwin, A.P. Leventis Conservation Foundation, AP Leventis Ornithological Research Institute, the British Ornithologists’ Union and the Linnean Society”

“This work was supported by Chris Goodwin, A.P. Leventis Conservation Foundation, AP Leventis Ornithological Research Institute, the British Ornithologists’ Union and the Linnean Society. This is paper number (to be completed at proof stage) from the AP Leventis Ornithological Research Institute”

“Chris Goodwin, A.P. Leventis Conservation Foundation, AP Leventis Ornithological Research Institute, the British Ornithologists’ Union and the Linnean Society”

6. We notice that your supplementary [figures/tables] are included in the manuscript file. Please remove them and upload them with the file type 'Supporting Information'. Please ensure that each Supporting Information file has a legend listed in the manuscript after the references list.

Reviewers' comments:

Reviewer's Responses to Questions

**Comments to the Author**

1. Is the manuscript technically sound, and do the data support the conclusions?

Reviewer #1: Yes

Reviewer #2: Partly

2. Has the statistical analysis been performed appropriately and rigorously? 

Reviewer #1: Yes

Reviewer #2: Yes

3. Have the authors made all data underlying the findings in their manuscript fully available?

Reviewer #1: No

Reviewer #2: No

4. Is the manuscript presented in an intelligible fashion and written in standard English?

Reviewer #1: Yes

Reviewer #2: Yes

5. Review Comments to the Author

Reviewer #1: Review – Plos One

Comparing migration of Whinchats Saxicola rubetra from Liberia and Nigeria: behaviours modified by differences in geography but unlikely to be constrained by them

In this article, the authors describe the migration strategies of Whinchat, tagging birds from two non-breeding sites, allowing an unprecedented comparison with an impressive sample size for such a small migrant. This article provides valuable insight into the migration of this species, but also very important data that can apply to the Afro-palearctic migratory system in general, provided that more research will be carried out using a similar design. This work is worth publishing in Plos One and will be of interest for the specialized readership as well as for readers interested in more general novelty in biological research.

I listed minor suggestions hereafter. The only major remark concerns the visual classification method (see under methods), and hereby the limitations of the light-level geolocation to classify migration legs. This should be better explained in the methods, including more precisions on the general uncertainties, and some caution should be taken while presenting migration leg speed for example.

Title: I would suggest rephrasing the title in a shorter, simpler and clearer way (the second part takes time to associate to its meaning, at least for me). Mention could be made in the title that the study is based on non-breeding sites, because the “from” could be misleading for the non-specialized readership and because this is what makes this study more interesting.

Abstract

L. 19: “whinchats” plural?

L. 19: rephrasing suggestion: “a declining palearctic-breeding passerine” for example.

L. 21: not clear what is meant with differences in migratory connectivity between the two sites (connectivity is the fact that two non-breeding populations would go to two distinct breeding areas?) maybe reframe and put the connectivity argument into one separate sentence?

L. 23: “distance, duration…” (not necessary to mention speed, as it is here derived from distance and duration, no data on real speed in the air is shown).

L. 24 not clear from the sentence in the abstract why these two populations should have a different number of stopover sites (and not sure this is important for the abstract).

L. 28: “significant loop migration” : to avoid confusion, I suggest omitting this unnecessary “significant” unless related so statistical analyses here

L. 35: “able to adapt to potential climate change across Europe and West Africa”: this conclusion is not well supported by the data; I would be more cautious with mentioning this like that here.

Introduction

L. 41: probably true. However, most of these birds eat insects and this should be mentioned as a potential cause of decline, if the topic of the cause of decline is approached (not necessarily the core topic of this paper). Furthermore, declining long-distance migrants that eat seeds are persecuted (ortolan, turtle doves) and also decline. So there is a broad spectrum of reasons why these birds decline (with some exceptions locally), but discussing this here might not be needed.

L. 46: “to such changes in the non-breeding area “?

L. 49: I disagree with this expectation as it is phrased (many species have different migration strategies across populations, see blackcap for example), but this might be related to the phrasing.

Methods:

L. 205-209: I have some doubts about the accuracy of visual inspection method, as whinchats migrate on a mostly north-south axis and at times close to equinox (especially for the post-breeding migration). Could you provide some more details and quantify uncertainties, even broadly? Are there situations where migration periods could not be assessed accurately? I would be surprised if not.

L. 289: it is likely that there were stopovers of less than two days, but they were not detected with the classification method. Stopovers of one day are very common in other species while crossing the desert and the Mediterranean (see for example https://doi.org/10.1186/s40462-023-00381-6, which refers to a closely related species that performs such stops of less than a day, for example). I expect such short stops to occur in Whinchats at least partly: could you give more details about this?

L.298-299: I might have missed it, but did you set a time threshold to consider a non-breeding site? There are often stop over sites located close to non-breeding sites, but with only a few days I would not consider them as alternative non-breeding sites but rather stopover sites within the non-breeding region.

Results:

L. 353: did you have locations on the sea, and did you use a sea mask to correct for this?

L. 354: is the spatial precision sufficient to qualify the country in which the bird was, or should you stay with more general areas/directions?

“The subsequent flight for this bird crossed from the southern coast of Ghana to the

Mediterranean coast of Tunisia in a single leg of 3,538 km over a 5-day period”

If referring to more precise migration data in other species (https://doi.org/10.1186/s40462-023-00381-6 for example), it is likely that this bird landed over the day at least some hours and did not fly non-stop (in which case it would likely have taken max. 3 days), this could be explained better and discussed in the discussion section. It would be awesome to fit pressure loggers on Whinchats in the future to refine this (could be mentioned in the discussion).

L. 414: for discussion: could this be related to the onset of rains in Nigeria in the northern spring, from which Whinchats could benefit to refuel before crossing the desert (while rainfalls are less strongly seasonal in Liberia and start earlier?)

L. 447-453: I suggest skipping or being more cautious about the estimate of migration speed, as it is likely biased by the classification method and not a very important finding of the study.

L. 625: this partly solves my previous remark, but I would make it clearer in methods and results that the speed you measure is including some stationary phases. Since this migration leg speed is not a key finding of the study, I would give it less importance overall in the paper.

L. 685: I find this example of extremely short total migration time very interesting, could it be better highlighted in the results and maybe even in the abstract?

L. 694 – 695: probably not true for some desert stops?

Figures

On figure 3 especially, but also other figures, a visual representation of uncertainties could be valuable, especially for supposedly southwards movements before migration.

Thanks for this very interesting read.

Reviewer #2: In this study, authors clarified the characteristics of migration strategies of the whinchat, comparing them with two non-breeding populations. Authors focused on the 5 main topics of the two populations, i.e., 1) breeding ranges, 2) secondary non-breeding sites, 3) migration distance, the number of stopover sites and loop migration, 4) migration duration and phenology and 5) migration leg and stopover duration. The large dataset of whinchat’s migration presented some convincing results, but some problems should be corrected. My comments are listed below.

L130-131　

I think that there is insufficient evidence to provide a hypothesis that loop migration does not occur in Liberian wintering population. Please provide additional information regarding the reasoning behind your hypothesis.

L193

How did you treat the geolocator data during the equinox period? Tracked individuals appear to depart or arrive during the equinox period, as shown in Table S1. However, there is no description of the data analysis for this period.

L252-267

This paragraph should be moved to the result section.

L276

‘therefore do not include a random effect.’

Here, please add a sentence such as ‘i.e., analysed using Generalized Linear Models’.

L277, L433-441

Model selection and statistical tests should not be conducted simultaneously because their objectives are different. I think that model selection based on AIC/AICc is unnecessary because the focus, as in the other analyses, was on the significance of the explanatory variable (including the interaction term).

L377

‘-.0’: is this correct?

L358-360

I could not understand how the distance was calculated. Please explain it more precisely.

L440

I think that P=0.11 is not marginal.

L568　

Figure 2? I think Figure 1 is correct here.

L568

‘Our Liberia results possibly show this trend as well’ I think that it is over discussion here.

L633-634

Replace [ ] with ( ).

L1010

I think that the line colors of the minimum convex polygons are reversed between the Liberian and Nigerian populations in Figure 2.

L1034　

It seems that the table is misaligned. Is it correct?

6. PLOS authors have the option to publish the peer review history of their article (what does this mean? ). If published, this will include your full peer review and any attached files.

**Do you want your identity to be public for this peer review?** For information about this choice, including consent withdrawal, please see our Privacy Policy .

Reviewer #1: No

Reviewer #2: No

---

## [Author Response · Author response to Decision Letter 1]

28 Mar 2025

PONE-D-25-07789

Comparing migration of Whinchats Saxicola rubetra from Liberia and Nigeria: behaviours modified by differences in geography but unlikely to be constrained by them

PLOS ONE

Our responses below:

Both reviewers agreed with the value of the dataset and were impressed with the sample size. They were, however, both unsure of some part of the methodology, in particular about handling of the geolocator data around the time of equinox and how you dealt with the uncertainty. Please clarify this, as well as responding to other suggestions made by the reviewers.

#Thank you for your prompt and helpful review. I have responded to all the referees’ comments fully and in detail as they suggested (in italics below), apart from a couple of places where the comments are subjective and less important. These are marked with NO CHANGE along with my argument to keep the text the same. None are deal breakers, simply differences in style of writing and I believe my approach is clearer in these cases.

#Done

#Done

#Done

#NA

#NA

Journal Requirements:

#Done

#Done although it was unclear how to allocate unequal authorship. Will Cresswell (me), designed and supervised the research, collected some of the data, analysed all the data, wrote the paper entirely; the rest of the team contributed to the bulk of the data collection and edited the paper.

“Chris Goodwin, A.P. Leventis Conservation Foundation, AP Leventis Ornithological Research Institute, the British Ornithologists’ Union and the Linnean Society”

#Please regard the below as our statement:

“This work was supported by Chris Goodwin, A.P. Leventis Conservation Foundation, AP Leventis Ornithological Research Institute, the British Ornithologists’ Union and the Linnean Society. This is paper number (to be completed at proof stage) from the AP Leventis Ornithological Research Institute”

Please remove any funding-related text from the manuscript

#Done

and let us know how you would like to update your Funding Statement. Currently, your Funding Statement reads as follows:

“Chris Goodwin, A.P. Leventis Conservation Foundation, AP Leventis Ornithological Research Institute, the British Ornithologists’ Union and the Linnean Society”

#Please regard the below as our statement of funding:

“Chris Goodwin, A.P. Leventis Conservation Foundation, AP Leventis Ornithological Research Institute, the British Ornithologists’ Union and the Linnean Society”

#Done

The data and R script files has been deposited in the University of St Andrews open access data depository PURE with a DOI of:

Pure ID: 316400564 (The St Andrews Data Depository)

Title: Comparing migration of Whinchats Saxicola rubetra from the non-breeding grounds in Liberia and Nigeria: differences due to geography but otherwise very similar (dataset)

Inactive DOI: https://doi.org/10.17630/59c8ff25-858d-4dcb-8f47-6b65894bd0e0

Once the manuscript is accepted and I have notified the University Research Depository office that the data files, in their final version, have been uploaded in Pure, they will proceed with the activation of the DOI.

Activation of the DOI will make the metadata for the dataset discoverable on the research portal. However, they will release the data files only once the article has been published online.

#Our ethics statement now appears at the end of the methods and nowhere else.

6. We notice that your supplementary [figures/tables] are included in the manuscript file. Please remove them and upload them with the file type 'Supporting Information'. Please ensure that each Supporting Information file has a legend listed in the manuscript after the references list.

#All supporting information is now removed from the main text apart from the legend list at the end of the manuscript.

A file containing Supporting Information only has been made and uploaded

Reviewers' comments:

Reviewer's Responses to Questions

Comments to the Author

1. Is the manuscript technically sound, and do the data support the conclusions?

Reviewer #1: Yes

Reviewer #2: Partly

2. Has the statistical analysis been performed appropriately and rigorously?

Reviewer #1: Yes

Reviewer #2: Yes

3. Have the authors made all data underlying the findings in their manuscript fully available?

Reviewer #1: No

Reviewer #2: No

DONE as above

The data and R script files has been deposited in the University of St Andrews open access data depository PURE with a DOI

4. Is the manuscript presented in an intelligible fashion and written in standard English?

Reviewer #1: Yes

Reviewer #2: Yes

5. Review Comments to the Author

Reviewer #1: Review – Plos One

Comparing migration of Whinchats Saxicola rubetra from Liberia and Nigeria: behaviours modified by differences in geography but unlikely to be constrained by them

In this article, the authors describe the migration strategies of Whinchat, tagging birds from two non-breeding sites, allowing an unprecedented comparison with an impressive sample size for such a small migrant. This article provides valuable insight into the migration of this species, but also very important data that can apply to the Afro-palearctic migratory system in general, provided that more research will be carried out using a similar design. This work is worth publishing in Plos One and will be of interest for the specialized readership as well as for readers interested in more general novelty in biological research.

I listed minor suggestions hereafter. The only major remark concerns the visual classification method (see under methods), and hereby the limitations of the light-level geolocation to classify migration legs. This should be better explained in the methods, including more precisions on the general uncertainties, and some caution should be taken while presenting migration leg speed for example.

#Corrections made as per Referee’s suggestions and are detailed below under the specific points

Title: I would suggest rephrasing the title in a shorter, simpler and clearer way (the second part takes time to associate to its meaning, at least for me). Mention could be made in the title that the study is based on non-breeding sites, because the “from” could be misleading for the non-specialized readership and because this is what makes this study more interesting.

#Changed as suggested to:

Comparing migration of Whinchats Saxicola rubetra from the non-breeding grounds in Liberia and Nigeria: differences due to geography but otherwise very similar

Abstract

L. 19: “whinchats” plural?

#Changed as suggested.

L. 19: rephrasing suggestion: “a declining palearctic-breeding passerine” for example.

#Changed as suggested

L. 21: not clear what is meant with differences in migratory connectivity between the two sites (connectivity is the fact that two non-breeding populations would go to two distinct breeding areas?) maybe reframe and put the connectivity argument into one separate sentence?

#Changed as suggested to:

We predicted differences, resulting from the geographical location of the two non-breeding sites, in location of respective breeding areas (migratory connectivity),

L. 23: “distance, duration…” (not necessary to mention speed, as it is here derived from distance and duration, no data on real speed in the air is shown).

#Deleted as suggested

L. 24 not clear from the sentence in the abstract why these two populations should have a different number of stopover sites (and not sure this is important for the abstract).

#NOT CHANGED

I disagree. Number of stop-overs is probably what really determines survival in long distance migrants as each reflects a site that must be suitable for the migration to succeed (the chain-link hypothesis). That Liberian populations have longer migrations and so more stop overs might mean that these populations are declining while Nigeria populations aren’t – so this is important.

L. 28: “significant loop migration” : to avoid confusion, I suggest omitting this unnecessary “significant” unless related so statistical analyses here

#Clarified as suggested.

“Liberian birds had longer migrations in distance and duration, and more stopovers, but only Nigerian birds showed a statistically significant difference in longitude comparing spring and autumn migrations (i.e. a clear loop migration).”

L. 35: “able to adapt to potential climate change across Europe and West Africa”: this conclusion is not well supported by the data; I would be more cautious with mentioning this like that here.

#Modified to be clearly speculative – but it is an important and I think probably correct point, even if our results are only concordant with this conclusion.

“This suggests whinchats are well adapted to the current variable geography and so may have the capacity to adapt to potential climate change across Europe and West Africa, although average quality and availability of stopover sites may be contributing to declines.”

Introduction

L. 41: probably true. However, most of these birds eat insects and this should be mentioned as a potential cause of decline, if the topic of the cause of decline is approached (not necessarily the core topic of this paper). Furthermore, declining long-distance migrants that eat seeds are persecuted (ortolan, turtle doves) and also decline. So there is a broad spectrum of reasons why these birds decline (with some exceptions locally), but discussing this here might not be needed.

#NO CHANGE MADE

I don’t think we need to go into the detail here, especially in the very general first paragraph, and anyway, broadly speaking, the referee seems ok with the original text.

L. 46: “to such changes in the non-breeding area “?

#Added as suggested

L. 49: I disagree with this expectation as it is phrased (many species have different migration strategies across populations, see blackcap for example), but this might be related to the phrasing.

#Rephrased as suggested:

“Some migration characteristics should be similar across all populations of a species”

Methods:

L. 205-209: I have some doubts about the accuracy of visual inspection method, as whinchats migrate on a mostly north-south axis and at times close to equinox (especially for the post-breeding migration). Could you provide some more details and quantify uncertainties, even broadly? Are there situations where migration periods could not be assessed accurately? I would be surprised if not.

#Additional detail added as suggested; in short sensitivity analysis indicates that the few equinoxial movements make no difference to the results.

Added text, one paragraph up from data analysis subheading, methods section:

“Geolocator locations are also subject to imprecision calculating latitude at the equinox periods (the few days either side of March 20th and September 22nd when regardless of latitude, daylength is similar). Although latitude is unreliabl

---

## [Decision Letter · Decision Letter 1]

21 Apr 2025

Comparing migration of Whinchats Saxicola rubetra from the non-breeding grounds in Liberia and Nigeria: differences due to geography but otherwise very similar

PONE-D-25-07789R1

Dear Dr. Cresswell,

We’re pleased to inform you that your manuscript has been judged scientifically suitable for publication and will be formally accepted for publication once it meets all outstanding technical requirements.

Kind regards,

Shoko Sugasawa

Academic Editor

PLOS ONE

Additional Editor Comments (optional):

Thank you very much for the quick revision of the manuscript. Both reviewers and I found the revision to be mostly sufficient. Please refer to and incorporate the remaining minor feedback from Reviewer 2, and I think the manuscript is ready for publication.

Reviewers' comments:

Reviewer's Responses to Questions

**Comments to the Author**

1. If the authors have adequately addressed your comments raised in a previous round of review and you feel that this manuscript is now acceptable for publication, you may indicate that here to bypass the “Comments to the Author” section, enter your conflict of interest statement in the “Confidential to Editor” section, and submit your "Accept" recommendation.

Reviewer #1: All comments have been addressed

Reviewer #2: (No Response)

2. Is the manuscript technically sound, and do the data support the conclusions?

Reviewer #1: Yes

Reviewer #2: Yes

3. Has the statistical analysis been performed appropriately and rigorously? 

Reviewer #1: Yes

Reviewer #2: Yes

4. Have the authors made all data underlying the findings in their manuscript fully available?

Reviewer #1: No

Reviewer #2: No

5. Is the manuscript presented in an intelligible fashion and written in standard English?

Reviewer #1: Yes

Reviewer #2: Yes

6. Review Comments to the Author

Reviewer #1: This version should be ready for publication. All points questioned in my previous review were addressed in the revised manuscript. In the few cases where the authors decided not to follow a suggestion, they justified their decision.

I agree with the author's response about sea mask bias.

Reviewer #2: I am approximately satisfied with your revision, but some problems are still present. Please check the comments below.

[L143] ‘Liberian and Nigerian whinchats will have similar:’ should be moved to next raw in hypothesis 5 as ‘5. Liberian and Nigerian whinchats will have similar: migration leg distance and ~~’

[L449] Add the “Liberian” and “Nigerian” at the top of the column for clarity of correspondence of the data and the non-breeding populations.

[L452] P , 0.0004 ->> P = 0.0004

[L542] I previously pointed out that "P = 0.11 is not marginal." You deleted the word "marginal" according to my suggestion, but it seemed to interpret the result as "significant". However, a P-value of 0.11 typically indicates non-significance. That said, the difference in mean migration leg distance between spring and autumn migrations of the Nigerian wintering population appears to be quite large (over 400 km), based on the values in Table 1. Please recheck accuracy of the results, i.e., mean values, standard errors, and t- and P-values (-488 km + 309SE, t = -1.5, P = 0.11), and revise the Results and Discussion sections (e.g., line 603, etc.) accordingly to reflect the accurate results.

7. PLOS authors have the option to publish the peer review history of their article (what does this mean? ). If published, this will include your full peer review and any attached files.

**Do you want your identity to be public for this peer review?** For information about this choice, including consent withdrawal, please see our Privacy Policy .

Reviewer #1: No

Reviewer #2: No

---

## [Editor Report · Acceptance letter]

PONE-D-25-07789R1

PLOS ONE

Dear Dr. Cresswell,

I'm pleased to inform you that your manuscript has been deemed suitable for publication in PLOS ONE. Congratulations! Your manuscript is now being handed over to our production team.

Kind regards,

on behalf of

Dr. Shoko Sugasawa

Academic Editor

PLOS ONE